# Establishment and Comprehensive Molecular Characterization of an Immortalized Glioblastoma Cell Line from a Brazilian Patient

**DOI:** 10.3390/ijms242115861

**Published:** 2023-11-01

**Authors:** Fernanda F. da Silva, Fernanda C. S. Lupinacci, Bruno D. S. Elias, Adriano O. Beserra, Paulo Sanematsu, Martin Roffe, Leslie D. Kulikowski, Felipe D’almeida Costa, Tiago G. Santos, Glaucia N. M. Hajj

**Affiliations:** 1International Research Center/CIPE, A.C. Camargo Cancer Center, São Paulo 01508-010, Brazil; fernandake25@gmail.com (F.F.d.S.); bruno.elias@accamargo.org.br (B.D.S.E.); tsantos@accamargo.org.br (T.G.S.); 2National Institute of Science and Technology in Oncogenomics (INCITO), São Paulo 01509-900, Brazil; 3Neurosurgery Department, A.C. Camargo Cancer Center, São Paulo 01509-010, Brazil; 4Children’s Hospital of Eastern Ontario Research Institute, Ottawa, ON K1H 8L1, Canada; 5Cytogenomics Laboratory, Hospital das Clínicas da Faculdade de Medicina, Universidade de São Paulo, São Paulo 05403-010, Brazil; lesliekulik@usp.br; 6Department of Anatomic Pathology, A.C. Camargo Cancer Center, São Paulo 01509-010, Brazil; felipe.costa@accamargo.org.br

**Keywords:** glioblastoma, cell lines, primary cilia, Akt, mTOR, copy number variation, methylation profile

## Abstract

Glioblastoma (GBM) is the most common and aggressive primary brain tumor in adults, with few effective treatment strategies. The research on the development of new treatments is often constrained by the limitations of preclinical models, which fail to accurately replicate the disease’s essential characteristics. Herein, we describe the obtention, molecular, and functional characterization of the GBM33 cell line. This cell line belongs to the GBM class according to the World Health Organization 2021 Classification of Central Nervous System Tumors, identified by methylation profiling. GBM33 expresses the astrocytic marker GFAP, as well as markers of neuronal origin commonly expressed in GBM cells, such as βIII-tubulin and neurofilament. Functional assays demonstrated an increased growth rate when compared to the U87 commercial cell line and a similar sensitivity to temozolamide. GBM33 cells retained response to serum starvation, with reduced growth and diminished activation of the Akt signaling pathway. Unlike LN-18 and LN-229 commercial cell lines, GBM33 is able to produce primary cilia upon serum starvation. In summary, the successful establishment and comprehensive characterization of this GBM cell line provide researchers with invaluable tools for studying GBM biology, identifying novel therapeutic targets, and evaluating the efficacy of potential treatments.

## 1. Introduction

Diffuse gliomas in adults are the most common malignant primary brain tumor. The most recent 2021 classification from the World Health Organization (WHO) divides these tumors into astrocytomas, oligodendrogliomas, and glioblastomas (GBMs) based on histologic features and the presence or absence of a somatic mutation in the *IDH1/2* genes [1]. The main change in the 2021 classification is that only tumors that are *IDH1/2* wild-type are included in the GBM group. GBMs also have an increase in chromosome 7, a loss of chromosome 10, mutations of the telomerase reverse transcriptase (*TERT*) promoter, amplification and mutation of the epidermal growth factor receptor (*EGFR*), and loss of cyclin-dependent kinase inhibitor 2A/B (*CDKN2A/B*). In addition, tumors that can be histologically classified as low-grade diffuse gliomas (i.e., without microvascular proliferation or necrosis) but have the *IDH1/2* wild-type and all molecular features of GBM (*EGFR* amplification, combined Chr7 gain and Chr10 loss, or *TERT* promoter mutation) are now classified as GBM [1].

The presence of an *IDH1* or *IDH2* mutation and an unbalanced translocation between chromosomes 1 and 19 (1p/19q-codeleted) classifies oligodendrogliomas, whereas an *IDH1/2* mutation without 1p/19q-codeletion defines astrocytomas [2].

GBMs are among the most lethal tumor types, with a poor prognosis of approximately 13 months survival. Treatments include surgery, radiotherapy, and chemotherapy with the alkylating agent temozolomide [3]. However, due to the aggressiveness of the disease, only 5% of patients diagnosed with GBM survive longer than 5 years [4]. Furthermore, resistance to treatment with alkylating drugs can be observed upon the presence of O-6-methylguanine-DNA methyltransferase, encoded by the *MGMT* gene, which repairs damaged guanine nucleotides, thus preventing DNA damage-associated cell death. *MGMT* copy number loss or methylation of the promoter reduces the expression of the enzyme and increases the efficacy of chemotherapeutic treatments [5].

Considering the high incidence and severity of GBMs, extensive research efforts have been devoted to unraveling the underlying molecular mechanisms of this tumor type. These studies have yielded valuable insights into the molecular biology of GBMs. Through the integrated analysis of chromosomal and genetic aberrations in GBMs, researchers have identified three crucial signaling pathways: receptor tyrosine kinase receptors (RTKs), p53, and RB [6]. For instance, frequent deletions of the tumor suppressor gene Phosphatase and Tensin Homolog (*PTEN*) occur as a result of chromosome 10 loss, which leads to dysregulated activation of the PI3K/Akt/mTORC1 pathway [6,7]. Despite the significant progress made in understanding the disease, the available treatment options remain limited, and targeted therapies have proven ineffective thus far [8].

Cancer cell lines play a crucial role as primary in vitro models for investigating fundamental aspects of cellular and molecular tumor biology. Moreover, they are invaluable tools for the preclinical testing of novel drugs and therapies, offering essential insights that can ultimately enhance patient outcomes. Consequently, there is a pressing need for cell models that accurately represent the characteristics of brain tumors. However, commonly used commercial GBM cell lines were established approximately four decades ago, and as these cells undergo repeated in vitro passages, they often acquire phenotypic traits and numerous genetic abnormalities that bear little resemblance to the original human tumor [9].

For instance, the commonly used GBM cell line, U-87MG, has been found to deviate from its original characteristics, as evident from differences in mutational profiles between the original tumor and the currently available cell populations [10]. Moreover, many established cell lines lack associated clinical data, impeding the establishment of clinical and molecular correlations. Furthermore, when injected into murine models, existing cell lines fail to recapitulate the tumor microenvironment observed in human GBM, including high infiltration and microvascular proliferation [11,12].

To address this challenge, it is crucial to enhance preclinical biological models by developing new primary and immortalized cell lines with well-defined molecular characteristics, enabling more advanced drug testing. In this study, we present the isolation of a primary cell culture derived from a GBM tumor (GBM33). Our investigation also encompasses a comprehensive characterization of this cell line, including genomic alterations, expression of marker proteins, morphological features, growth kinetics, and activation of cell signaling pathways.

## 2. Results

### 2.1. Patient and Tumor Data

Patient GBM33 was a male diagnosed at the age of 76. Magnetic Resonance Imaging (MRI) studies indicated the presence of a mass in the right frontal lobe (Figure 1A). The patient experienced gait alteration and dysarthria and underwent partial surgical removal of the tumor in 2016, followed by radiotherapy and temozolomide treatment. Immunohistochemistry analysis revealed no staining for synaptophysin orGlial Fibrillary Acidic Protein (GFAP), positive staining for p53, and a Ki-67 labeling index of 70%. Disease progression occurred 759 days after surgery, and the patient’s overall survival from the date of surgery was 1001 days.

Genomic characterization of the tumors was performed through Copy Number Variant (CNV) and methylation analysis. GBM33 displays monosomy of Chromosomes 9/10/19, which includes *PTEN* and *MGMT*, and gains in Chromosomes 16/20, which involved amplification of *PDGFRA*, *CDK4*, and *MDM2* (log2 value above 0.4) (Figure 1B). The remaining copy of *MGMT* presented promoter methylation, suggesting a complete absence of expression of this gene. GBM33 exhibited wild-type *IDH1/2* status. Methylation analysis indicated that the tumor belonged to the GBM class, as defined by the Heidelberg and National Cancer Institute (NCI) classifiers [13] (Figure 1C). In whole exome sequence, GBM33 presented 176 Single Nucleotide Variants (SNV) (Table 1). From these, 95 variants were novel, and 81 were described alterations. No variants were found in frequently mutated genes in GBM (Appendix A).

We further filtered newly identified variants of potential interest that could play a role as new markers for personalized therapy of GBM (Table 2). We identified 16 novel variants, including a *PDGFRA* variant p.P441S.

### 2.2. Isolation and Characterization of GBM33 Cell Line

Fresh tumor fragments from the GBM33 patient were obtained during surgery. The tumors were subjected to trypsin digestion followed by mechanical dissociation and cultured in DMEM supplemented with 20% FCS. GBM33 cells could be successfully cultured, although they displayed slow growth for the first three months. After the initial period, the growth rate of GBM33 cells increased, and serum concentration was reduced to 10%. Throughout the culture process, primary cultures initially exhibited different cell morphologies, including fibroblasts, neurons, and astrocytes. However, after successive passages, only cells with astrocytic morphology remained (Figure 2A). The cell line was passaged for more than one hundred generations.

GBM33 was initially characterized for glial and neuronal markers, exhibiting staining for the neuronal markers β-III tubulin (Figure 2B) and neurofilament protein (NFP) (Figure 2C) and, with low staining for the astrocytic marker, GFAP (Figure 2D).

Genomic characterization was performed by exon sequencing followed by variant calling, which revealed a total of 34 additional variations (3 indel and 31 missense SNVs) between the cells and their respective original tumor. The identified variants are described in Appendix A. CVN profiles were also determined in comparison with the original tumor (Figure 2E,F). Cell line and original tumor share the majority of CNV alterations, including losses in Chr 9/10 that included *PTEN* and *MGMT* and gains in Chr 20, which included amplification of *PDGFRA*, *CDK4*, and *MDM2*. In comparison, commercial line U-87MG present partial losses in Chr 6pq/9p/11q/12q/13q/14q and 20p and partial gains in Chr 7/13q and 20q; LN-18 cells present gains in Chr 7/16/18p/20q and losses in Chr 10p/; and LN-229 cells present gains in Chr 1q/3p/5/7/9p/12q/20/22q and losses in Chr 6/10/11/12q/13/14/18 and 19.

### 2.3. Tumorigenic Properties of GBM33

We compared the proliferation of our newly established GBM cell line with the commercially available U-87MG cell line. GBM33 presented growth rates higher than U-87MG in the presence of Fetal Calf Serum (FCS), while under serum starvation, all cells showed growth arrest (Figure 3A). The resistance of the cells to temozolamide treatment was evaluated by 3-(4,5-dimethylthiazol-2-yl)-2,5-diphenyltetrazolium bromide (MTT), and GBM33 exhibited a very similar sensitivity, comparable to U-87MG (Figure 3B). An important characteristic of tumorigenic cells is their ability to grow independently of anchorage [15]. However, in the soft agar assay, GBM33 did not form colonies (Figure 3C).

Considering the loss of *PTEN* in GBM33, we evaluated the activation of proteins in the PI3K/mTOR pathway. Under serum starvation, GBM33 showed a significant reduction in Akt phosphorylation compared to cells treated with FCS. In contrast, the commercial cell lines LN-18 and U-87MG displayed baseline Akt phosphorylation even under starvation conditions (Figure 3D). Phosphorylation of p70S6K was observed in all cell lines in the absence of serum. Upon FCS treatment, LN-18 and GBM33 showed increased phosphorylation, while U87-MG did not exhibit a significant increase. Phosphorylation of rpS6 was observed in all cell lines under serum starvation, with a slight increase upon FCS treatment for all cell lines (Figure 3D).

The primary cilium (PC) is a specialized organelle involved in cell signaling [16]. In cancer, altered ciliogenesis has been observed in a variety of tumor types [17], including breast, prostate [18], renal [19], and nervous system. In particular, in a subset of Sonic hedgehog (SHH)-dependent medulloblastomas, the presence of PC is necessary for cancer maintenance [20].

In GBM, the PC has been associated with the proliferation and resistance to temozolomide [21,22,23,24,25]. However, previous studies mainly used primary cell cultures, as most commercial GBM cell lines either fail to induce ciliogenesis or have short cilia with a low percentage of ciliated cells [26,27,28,29]. Therefore, we examined whether GBM33 could generate cilia under serum starvation conditions. GBM33 was capable of producing cilia (Figure 4A), unlike LN-18 and LN-229 (Figure 4B,C), demonstrating that GBM33 retains the ability to modulate PC. In conclusion, our results demonstrate the establishment of a novel GBM cell line that can serve as an improved model for studying this deadly disease.

## 3. Discussion

The research focused on developing new treatments for GBMs is often constrained by the limitations of available preclinical models, which frequently fail to accurately replicate the disease’s essential characteristics.

To address these challenges, laboratories have developed various alternative models, including primary cell cultures [9,28,29], three-dimensional cultures such as oncospheres [11,30], and Patient Derived Xenografts (PDX) [31]. While these models offer the advantage of closely resembling the original tumor and retaining more of its characteristics, cell lines possess unique advantages: they are highly cost-effective, user-friendly, and provide an unlimited supply of material. They also circumvent ethical concerns associated with the use of animal and human tissue, while offering a homogeneous population of cells essential for consistent samples and reproducible results. Furthermore, cell lines allow for genetic manipulation, enabling the alteration of cellular functions. Consequently, cell lines are recommended for large-scale and preliminary drug screening, with further validation performed using primary cells or PDX [32].

In response to the need for reliable cell models, independent laboratories have generated self-established cell lines [9,33,34]. However, it is unfortunate that many of these cell lines often lack comprehensive characterization in terms of genetics, phenotypic traits, and immunocytochemical profiles. In this study, we address this gap by describing the obtention of a GBM cell line and thorough characterization of genomic alterations and cancer-related phenotypes. Our results demonstrate that these cell lines harbor genomic profiles similar to their corresponding matched tumors, suggesting their potential as reliable models for preclinical studies. Moreover, our newly developed cell lines have been classified according to the 2021 WHO Classification of Tumors of the Central Nervous System, thus representing a molecularly characterized group.

The CNV and methylation analyses conducted on our case confirmed its classification as GBM according to the latest WHO criteria. These analyses revealed characteristic genetic alterations, such as the gain of chromosome 7, loss of chromosome 10, mutations in *TERT* promoter, amplification and mutations in *EGFR*, and loss of *CDKN2A/B*. The majority of the CNV alterations were maintained in the GBM33 cell line when compared to the original tumor. However, we did observe minor alterations unique to the cell culture or the frozen tumor tissue, which can be attributed to the presence of non-tumoral cells in the original tumor that may influence the sequencing results [29].

We also characterized the phenotypic properties of the GBM cell line. The cell line exhibited a typical morphology with amorphous cell bodies and protrusions. Immunocytochemical characterization involves the use of antibodies against glial and neuronal markers, such as GFAP, β-III tubulin, and NFP. GFAP, an astrocyte marker used in GBM clinical diagnosis, was observed in all three cell lines with varying staining intensity. Neuronal markers β-III tubulin and NFP were also detected in the GBM cell lines, although their aberrant expression has been documented in patient biopsies and cell lines. Increased expression of β-III tubulin has been associated with higher malignancy in astrocytomas [9,35,36]. 

In terms of proliferation rates, the GBM33 demonstrated faster growth when compared to the commercial cell line U-87MG, and all cell lines exhibited reduced proliferation under serum starvation conditions. GBM33 also displayed a very similar resistance to temozolamide when compared to U-87MG; however, GBM33 did not demonstrate the ability to form colonies in soft agar.

PCs, also known as non-motile cilia, are specialized structures growing from the plasma membrane that play a role in cellular signaling and polarity. PCs have been linked to the Sonic hedgehog (Shh) signaling pathway, and defects in these structures underlie a variety of human diseases associated with central nervous system development problems [16].

In cancer, PC provides a spatially localized platform for oncogenic signaling pathways such as the Hedgehog, Notch, *WNT*, and receptor tyrosine kinase pathways [37]. In GBM, aberrant ciliogenesis has been linked to malignant phenotypes, including resistance to temozolamide treatment and cell proliferation [21,22,23,24,25]. Interestingly, impaired ciliogenesis has been observed as a response to oncogenic Ras activation in pancreatic tumors [38], further highlighting the importance of this structure to tumorigenic properties.

However, cilia studies are limited by the use of primary cultures, as reports suggest that commercial GBM lines generally lack cilia [26,27,39,40]. In contrast, the GBM33 cell line generated in this work was able to produce cilia in the absence of serum and exhibited a behavior similar to that of primary cultured cells.

The inability to develop a functional PC can have significant implications. It can impair the cell’s capacity to accurately perceive and adapt to its surroundings, affecting its ability to sense and respond to environmental cues. Additionally, it can disrupt the regulation of cell cycle progression, potentially leading to uncontrolled cell growth. Hence, the compromised ability of astrocytoma/glioblastoma cells to communicate and exhibit unrestrained growth may, in part, be attributed to underlying defects in the ciliogenesis process [25]. Thus, a cell model that retains the ability to grow PC is of pivotal importance for the studies of cell signaling pathways and drug response.

The PI3K/Akt/mTOR pathway is commonly activated in GBM through various mechanisms. Genetic alterations, such as amplification or mutation of receptor tyrosine kinases (RTKs) like *EGFR* and *PDGFR*, can lead to aberrant activation of PI3K. Additionally, loss of the tumor suppressor *PTEN*, a negative regulator of PI3K, is frequently observed in GBM, further enhancing pathway activation. Dysregulation of the PI3K/Akt/mTOR pathway in GBM cells contributes to tumorigenesis and treatment resistance, promoting cell survival, proliferation, and evading apoptosis [6,12]. Thus, targeting this pathway represents an attractive target for therapeutic intervention. Notably, when compared to commonly used commercial cell lines, GBM33 displayed inherent lower activation levels of Akt and p70S6K under serum starvation conditions. However, the addition of serum restored their activation, indicating that these cell lines serve as valuable models for investigating the intricacies of this pathway. By utilizing these cell lines, researchers can gain valuable insights into the regulation and potential therapeutic targeting of the PI3K/Akt/mTOR pathway in GBM.

In conclusion, our newly developed and characterized cell line may serve as invaluable tools for researchers, enabling them to investigate the intricate molecular mechanisms underlying GBM’s pathogenesis, identify potential therapeutic targets, and evaluate the efficacy of novel treatment strategies. Since the original patient characteristics and classification of our cells are available, this may facilitate the development of personalized medicine approaches, allowing for tailored treatments based on the genetic and phenotypic characteristics of individual patients’ tumors.

## 4. Materials and Methods

### 4.1. Commercially Available Cells Lines

U-87MG (ATCC^®^ HTB-14), LN-18 (ATCC^®^ CRL-2610), LN-229 (ATCC CRL-2611) cell lines were grown and maintained in Dulbecco’s Modified Eagle Medium (DMEM) supplemented with 10% fetal bovine serum (FCS) (Thermo Fisher Scientific, Waltham, MA, USA), 1% sodium pyruvate (Gibco^®^ by Life Technologies, Billings, MT, USA), 40 μg/mL gentamycin sulfate (Hipolabor Farmacêutica, Rio De Janeiro, Brazil), and maintained in humidified incubator with 5% CO_2_ at 37 °C. Cells were maintained in standard culture flasks (Thermo Fisher Scientific, Waltham, MA, USA) and replated upon reaching 80% of confluence.

### 4.2. Primary Cell Culture

We obtained one tumor fragment from a patient diagnosed with GBM at the AC Camargo Cancer Center in Sao Paulo, Brazil. The acquisition of the fragment occurred during patient curative surgery. The patients received no chemotherapy or radiotherapy treatment prior to surgery. The project was conducted with informed consent from the patients, and the study was approved by the Institutional Ethics Committee (Approval No. 082/18).

Fresh GBM fragments were carefully washed and minced in PBS. Subsequently, enzymatic dissociation was performed using 1 ml of Trypsin-EDTA (Thermo Fisher Scientific, Waltham, MA, USA) at 37 °C for 20 min. The isolated cells were then resuspended in DMEM supplemented with 20% FCS (Thermo Fisher Scientific, Waltham, MA, USA), 1% sodium pyruvate (Thermo Fisher Scientific, Waltham, MA, USA), and 40 μg/mL garamycin (Hipolabor Farmacêutica, Rio De Janeiro, Brazil). These cells were maintained in a humidified incubator with 5% CO_2_ at 37 °C. When the cells reached 80% confluency, they were passaged using 0.25% trypsin digestion.

### 4.3. Immunofluorescence Assay

Cells were seeded on coverslips and fixed with 4% paraformaldehyde (methanol-free) at room temperature. Following fixation, cells were permeabilized with PBS, 5% bovine serum albumin (BSA), and 0.5% Triton X-100. Primary antibodies were then added, including anti-neurofilament-200 (1:50, Sigma, n4142, Saint Louis, MI, USA), anti-GFAP (1:20, Cell Signaling, Danvers, MA, USA, #36705), anti-βIII-Tubulin (1:50, BD, New Jersey, NJ, USA, 560338), anti-acetyl-α-tubulin (1:100, Cell Signaling Cell Signaling, Danvers, MA, USA). The primary antibody incubation was carried out at room temperature for 1 h. Cells were washed 3 times with PBS and secondary antibodies, including anti-mouse Alexa Fluor^®^ 488 (1:500, Invitrogen, A11001, Waltham, MA, USA), anti-rabbit Alexa Fluor^®^ 488 (1:1000, Thermo Fisher Scientific, Waltham, MA, USA A11008), and DRAQ5 nuclear staining (1:250, Thermo Fisher Scientific, Waltham, MA, USA, 62251) were incubated at room temperature for 1 h. The slides were analyzed by fluorescence confocal microscopy (Leica Microsystems, Wetzlar, Germany).

### 4.4. Soft Agar Colony Formation Assay

A lower coating layer was prepared in 35 mm plates (Corning, NY, USA) with 0.5% UltraPureTM agarose (Thermo Fisher Scientific, Waltham, MA, USA, 15510-027) in 1× DMEM supplemented with 10% FCS. A total of 10^4^ cells were then suspended in an upper layer composed of 0.3% agarose in 1× DMEM with 10% FCS. The cells were maintained in a humidified chamber at 37 °C and 5% CO_2_. After 28 days, the colonies were stained with Nitro Blue Tetrazolium (Sigma Merck KGaA, Darmstadt, Germany, N-6876) in PBS in a humidified incubator with 5% CO_2_ at 37 °C overnight. Protocol adapted from (Borowicz et al., 2014) [15].

### 4.5. Cell Proliferation and Viability Analysis

A total of 3 × 10^3^ cells were seeded, and their cell numbers were recorded daily using a Neubauer chamber. Cell viability was determined by the MTT colorimetric assay as follows: 3 × 10^3^ cells were seeded in 24-well plates and, after 12 h, treated with temozolamide at concentrations of 10, 50, 100, and 150 µM. After 72 h at 37 °C, 0.5 ml/mL MTT dye [3-(4,5)edimethylthiahiazo-(-z-y1)-3,5-diphenytetrazolium bromide] (Sigma Merck KGaA, Darmstadt, Germany) was added to the medium, followed by incubation for 3 h. Cells were lysed with acid isopropanol, and absorbance was measured at a wavelength of 595 nm. The values were normalized by dividing the mean of all data points in each experiment. Experiments were made in triplicates.

### 4.6. Western Blot

Cells were plated and subjected to serum starvation for 48 h, followed by treatment with 10% FCS for two hours. Afterward, the cells were lysed using RIPA buffer, and the cell extracts were subjected to SDS-PAGE. Western blot analysis was performed using the following primary antibodies: AKT (1:1000, Cell Signaling, 9272, Danvers, MA, USA), phosphorylated AKT at Thr308 (1:1000, Cell Signaling, 2965 Danvers, MA, USA), phosphorylated AKT at Ser473 (1:1000, Cell Signaling, 9271S Danvers, MA, USA), rpS6 (1:5000, Santa Cruz, CA, USA, sc-74459), phosphorylated rpS6 at Ser235/236 (1:1000, Cell Signaling, 4856 Danvers, MA, USA), p70S6K (1:1000, Cell Signaling, 27085 Danvers, MA, USA), phosphorylated p70S6K at Thr389 (1:500, Cell Signaling, 9234S Danvers, MA, USA), anti-β-Actin (1:5000, Sigma Merck KGaA, Darmstadt, Germany, A5441), and PTEN (1:1000, Cell Signaling, 9188 Danvers, MA, USA). Secondary antibodies, either anti-mouse or anti-rabbit (1:1000, Thermo Fisher Scientific, Waltham, MA, USA), were utilized, and the protein bands were visualized using an ECL kit (Thermo Fisher Scientific, Waltham, MA, USA). Densitometry of bands was performed in Image J (National Institutes of Health, Bethesda, MD, USA) gel tools.

### 4.7. Copy Number Variation (CNV) Analysis Using Oncoscan

GBM33 tumor tissue and its derived cell line were analyzed using the OncoScan^®^ CNV Array (Affymetrix, Waltham, MA, USA) following manufacturer’s instructions. Briefly, DNA was extracted from frozen tumor samples and cultured cells using the Wizard^®^ HMW DNA Extraction Kit (Promega, Madison, WI, USA). The extracted DNA was then subjected to CNV and somatic mutation probe hybridization at 58 °C overnight (16–18 h). Each sample was divided into two wells, and the gap fill reaction was performed by adding dATP (A) and dTTP (T) (A/T) in one well and dGTP (G) and dCTP (C) (G/C) to the other well. Uncircularized probes and genomic DNA were digested by using a cocktail of exonucleases. The circular probes were subsequently linearized using a cleavage enzyme and amplified by PCR. Following a second round of PCR amplification, the 120 bp amplicons were cleaved and hybridized onto the OncoScan arrays for 16–18 h. The arrays were then stained and washed using the GeneChip^®^ Fluidics Station 450 and scanned in GeneChip^®^ Scanner 3000 7G (Affymetrix Waltham, MA, USA) to generate array images in DAT format. Array fluorescence intensity (CEL) files were automatically generated from DAT files using the Affymetrix^®^ GeneChip^®^ Command Console^®^ (AGCC). Then, CEL files containing the raw data were processed via the Affymetrix OncoScan Console (Thermo Fisher Scientific, Waltham, MA, USA).

### 4.8. Whole Exome Sequencing (WES) Analysis

Whole exome sequencing (WES) was performed on GBM33 tumor tissue and derived cell lines to identify SNVs. The sequencing was conducted by Macrogen (Seoul, Korea) using the Novaeq platform following the 150 bp paired-end sequencing protocol and SureSelect V6-post library.

Data analysis was carried out by the Bioinformatics Facility at the A. C. Camargo Cancer Center, utilizing the Genome Analysis Toolkit (GATK) pipeline for somatic variant analysis. The analysis involved sequence alignment, pre-processing, and variant calling. The alignment of FASTQ files to the reference human genome (version hg19/GRCh37) was performed using BWA software (version 0.7.17-r1188) in MEM mode. After alignment, the BAM files underwent pre-processing steps using GATK software (version 3.8), including MarkDuplicates, BaseRecalibrator, and PrintReads, with the number of threads set to 8. For PDX tumor sequencing data, the XenofilteR package was initially employed to exclude murine reads and generate a BAM file containing sequences aligned with the human genome. GATK pre-processing steps were not applied to these samples to preserve critical information for XenofilteR analysis. Variant calling was performed using GATK3.8 mutect2 software, following GATK best practices. The resulting VCF files were annotated using snpEff v4.3t and SnpSift v4.3t software. To eliminate polymorphisms, hard-filters were applied to the VCFs based on various databases, such as ExAC, gnomAD, and 1000G. For further analysis, the filtered VCF files were converted to MAF format using the maftools software in R 3.5. The potential variant effect on protein function was evaluated by the prediction tools SIFT and PolyPhen.

For the identification of novel variants of potential interest, the data were subjected to a more detailed analysis. Variants were filtered by presence in more than 20 reads, allele frequency greater than 5%, type of change missense, or that lead to loss of function and predicted to be pathogenic or possibly pathogenic by at least one prediction tool.

### 4.9. Methylation Analysis

GBM33 tumor tissue was subject to DNA methylation analysis using the Infinium Methylation EPIC BeadChip 850K (Illumina, San Diego, CA, USA) following the manufacturer’s instructions. Initially, DNA was extracted from frozen tumor samples using the Wizard^®^ HMW DNA Extraction Kit (Promega Madison, WI, USA). The extracted DNA underwent bisulfite conversion, followed by whole-genome amplification and enzymatic fragmentation. Samples were then hybridized to the beadchips. Single-base extension reactions using allele-specific primers and labeled dNTPs were performed. Chips were scanned to measure the intensities of the unmethylated and methylated beads. This generated two IDAT data files that were analyzed using the Brain Tumor Classifier developed by the University of Heidelberg, available online at molecularneuropathology.org (accessed on 23 October 2023) (Capper et al., 2018) [13]. This classifier utilizes machine learning and bioinformatics algorithms to provide CNV data, a calibrated score indicating the most likely methylation class, and an estimate of the methylation status of the *MGMT* gene promoter.

## Figures and Tables

**Figure 1 ijms-24-15861-f001:**
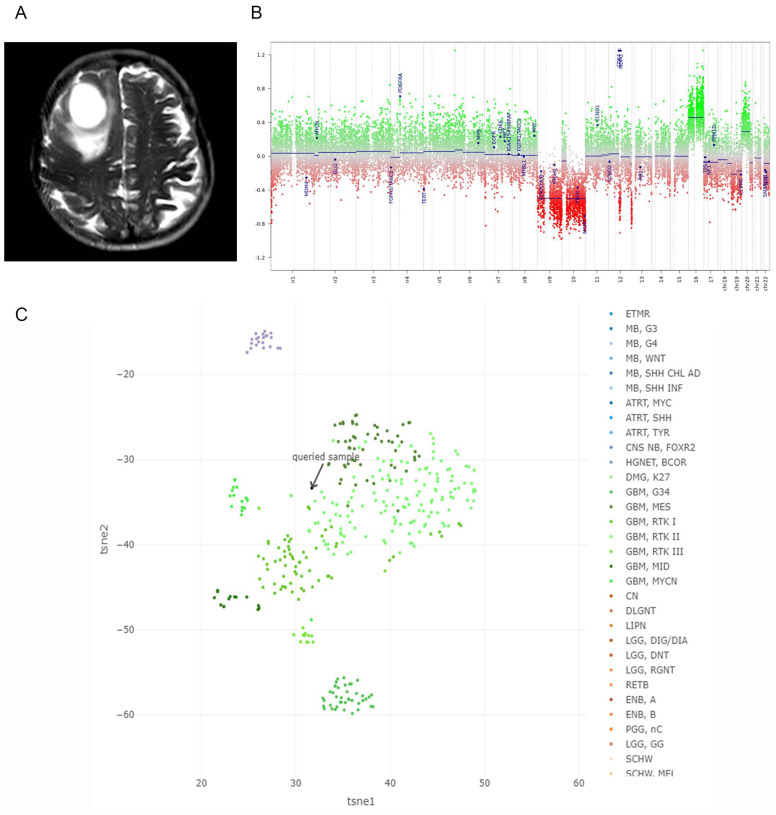
(**A**) MRI scans of GBM33 patient showing initial tumor size and localization. (**B**) Genomic characterization of GBM33 tumor. Copy Number Variant (CNV) plots calculated from DNA methylation array data. Depiction of Chromosome 1–22 with the p-arm (left) and the q-arm (right) separated by a dotted line. Gains/amplifications represent positive and losses negative deviations from the baseline. The probes of the array are combined in 8000 bins (green or red dots). (**C**) Classification of GBM33 in methylation-based classes [13].

**Figure 2 ijms-24-15861-f002:**
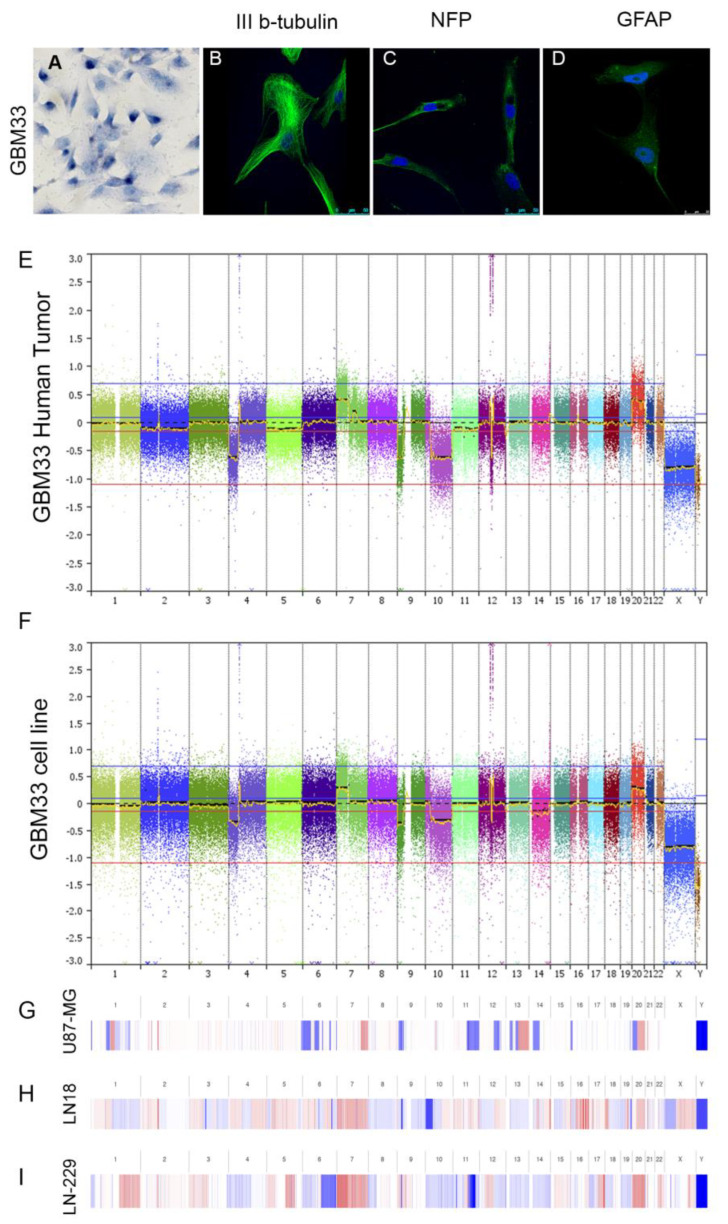
Cell morphology of GBM33 (**A**). GBM33 cells were subject to immunofluorescence assays against β-III tubulin (**B**), neuron filament protein (NFP) (**C**), and glial fibrillary acidic protein (GFAP) (**D**). (**E**–**I**) Genomic comparison of GBM33 original tumor (**E**), derived cell line (**F**), and commercial cell lines U87-MG (**G**), LN-18 (**H**), and LN-229 (**I**). (**E**,**F**) CNV plots calculated from DNA Oncoscan array data. (**G**–**I**) CNV plots generated in cBioPortal from the Cancer Cell Line Encyclopedia [14].

**Figure 3 ijms-24-15861-f003:**
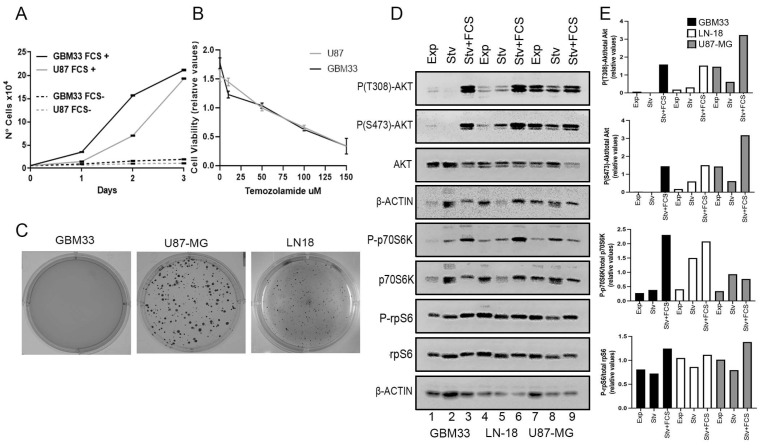
Growth and resistance properties of GBM33 cell line. (**A**) Growth curve of GBM33 in presence or absence of 10% FCS. (**B**) Cell viability in the presence of increasing concentrations of temozolamide measured by MTT. U87 cells were used for comparison. Results of three independent experiments, mean ± standard deviation. (**C**) Soft agar assay of GBM33, U87, and LN-18 cells. (**D**) Phosphorylation of proteins in the Akt/mTOR pathway. GBM33 (lanes 1–3), LN-18 (lanes 4–6), and U87-MG (lanes 7–9) cells were grown exponentially (Exp, lanes 1, 4, and 7) serum starved for 48 h (Stv, lanes 2, 5, and 8) or starved, followed by treatment with 10% FCS for 2 h (Stv + FCS, lanes 3, 6, and 9). Cell extracts were subject to Western blot against P-(T308)-AKT, P-(S473)-AKT, total AKT, P-(S389)-p70S6K, total p70S6K, P-(235/6)rpS6, total rpS6, and β-actin as loading control. (**E**) Densitometry plots of experiment in (**C**). Bands were quantified by densitometry and plotted as a ratio of the phosphorylated/total.

**Figure 4 ijms-24-15861-f004:**
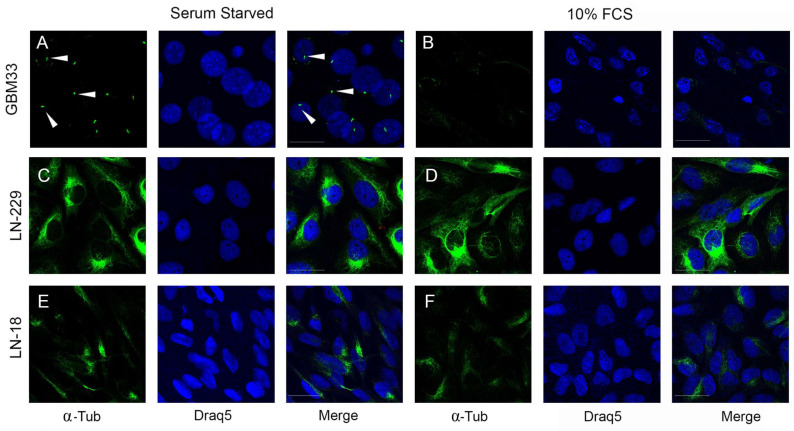
GBM33 cell line displaying primary cilia growth. GBM33 (**A**,**B**), LN-229 (**C**,**D**), and LN-18 (**E**,**F**) were maintained in 10% FCS (**B**,**D**,**F**) or serum starved for 24 h (**A**,**C**,**E**). Cells were fixed, and immunofluorescence against acetylated-α-tubulin (α-Tub—green) and nuclei staining (Draq5—blue) were performed. Bars, 20 µm. Arrowheads indicate primary cilia.

**Table 1 ijms-24-15861-t001:** Number of variants present in GBM33 tumor.

	GBM33		
Variant Classification	Novel	Annotated	Total
Frame_Shift_Del		1	1
Frame_Shift_Ins			
In_Frame_Del	6		6
Missense_Mutation	84	76	160
Nonsense_Mutation	5	4	9
Splice_Site			
Total Geral	95	81	176

**Table 2 ijms-24-15861-t002:** Novel variants of potential interest.

Gene Symbol	Variant	VAF	SIFT	PolyPhen
*PDGFRA*	p.P441S	0.856	3	3
*PCDHGA3*	p.L705P	0.448	2	3
*ECEL1*	p.P34H	0.506	1	2
*GRM6*	p.A161T	0.504	2	3
*CREBBP*	p.I1600M	0.483	2	
*NTF3*	p.K150Q	0.455	3	2
*NLRC3*	p.N183D	0.425	3	3
*TACC2*	p.D69Y	0.830	3	2
*PRODH2*	p.E428G	0.597	3	3
*DSP*	p.D1471A	0.389	3	1
*ULK3*	p.L414V	0.250	3	2
*FAM78B*	p.T147S	0.245	3	3

VAF—variant allele frequency, 3—deleterious (SIFT) or probably damaging (PolyPhen), 2—deleterious low confidence (SIFT) or possibly damaging (PolyPhen), T—tolerated, 1—tolerated low confidence (SIFT) or benign (Polyphen).

## Data Availability

Data is contained within the article or Appendix A. The data presented in this study are available in Appendix A.

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
