# Peer review of "Establishment and Comprehensive Molecular Characterization of an Immortalized Glioblastoma Cell Line from a Brazilian Patient"

_ijms, 2023, doi:10.3390/ijms242115861_

Round 1
Reviewer 1 Report
Comments and Suggestions for Authors
Author Response
We would like to acknowledge the manuscript reviewers for their generous allocation of time and their insightful, constructive comments. Their evaluation and feedback contributed to the refinement of this work, improving the quality and clarity of our manuscript.
Please find bellow a point by point answer to the reviewer's comments:
- Figure 2 A-D:the authors conduct an immunofluorescence assay to examine the presence of neuronal and astrocytic markers in the GBM33 line. Considering that GBM is classified as a grade IV astrocytoma, it raises questions about the relatively lower expression of GFAP. An explanation for this observation is needed.
Several studies have documented that GFAP expression is frequently diminished in human glioma specimens with increasing degree of anaplasia. In glial tumors, GFAP expression is frequently lost with increasing grade of malignancy, suggesting that GFAP is important for maintaining glial cell morphology or regulating astrocytoma cell growth. Most permanent human glioma cell lines are GFAP negative by immunocytochemistry. (10.1093/neuonc/noq145).
In addition, as discussed further in Reviewer 2 - Question 1, the original tumor presented no GFAP expression measured by routine clinical IHC, suggesting that low levels of GFAP found in the cells are compatible with low levels observed in the original tumor.
- The well-established characteristics of GBM, such as its high heterogeneity, aggressive invasiveness, and tendency to recur, have often been attributed to the presence of glioma stem-like cells (GSCs) within the tumour mass. Nevertheless, the GBM33 cell line appears to lack the capability to generate colonies in the soft agar assay as depicted in Figure 3C. This issue requires further investigation and analysis.
We agree with the reviewer that the importance of GSCs has been increasingly highlighted in the literature. However, the GSC population remains challenging both to define empirically and treat. In addition, regulation of the GSC in vitro has depended on multiple molecular mechanisms, such as epigenetics, cellular states, microenvironment, and other factors that potentially allow the GSC population to transition between SC and non-SC states. In this matter, we believe that a comprehensive study of the properties of GBM33 cell line regarding its stemness capabilities would require a lot of experimental standardization that are beyond the scope of this study.
- In lines 159-160, the authors mention their examination of the PI3K-mTOR pathway in response to the PTEN loss in GBM33line. In Figure 3D, they present the expression levels of p-AKT, p-P70, and p-RPS6 proteins to illustrate the up-regulation of the PI3K/Akt/mTOR signaling pathway as a result of PTEN loss. However, there are some ambiguities in the Western blot analysis: what exactly is the culture condition loaded in lanes 1-4-7 of the gel? Why is mTOR not included in the analysis? Additionally, it would be beneficial to include band densitometry data for clarity and quantification.
We apologize for the ambiguity in figure 3D. Lanes 1-4 and 7 represent exponentially growing cells without any treatment. We modified the figure and legend to make it more clear. We also performed band densitometry of the experiment and included it figure 3E. mTOR was not included in the analysis since p70S6K and rpS6 are mTORC1 targets, and their phosphorylation status is generally accepted in the literature as readout of mTOR activity.
- The noteworthy capability of the GBM33 line to form primary cilia under starvation conditions, as demonstrated in Figure 4, deserves more comprehensive exploration by the authors, both in the results and discussion sections. Additionally, exploring into the investigation of SHH, Notch, and WNT pathways could further enhance the significance of this study.
We agree with the reviewer and improved both the results and discussion on primary cilia.
Reviewer 2 Report
Comments and Suggestions for Authors
In their manuscript, da Silva et al report generation of one cell line from the tumor tissue of a patient with GBM. They performed a fair amount of molecular characterization and some characterization at the cellular level of the cell line and conclude that “…the successful establishment and comprehensive characterization of this GBM cell line provide researchers with invaluable tools for studying GBM biology, identifying novel therapeutic targets, and evaluating the efficacy of potential treatments.”
It is true the authors have created a GBM cell line, but they do not present any experimental support to their conclusion. Consequently, their conclusion, appears to be POTENTIAL implication at best.
Specific comments:
1. Line 93, it says, “…no staining for synaptophysin or GFAP”., whereas in line 125, it says “…low staining for the astrocytic marker GFAP”. Please explain why the tumor was negative for GFP whereas a cell line derived from it expressed GFAP at low level.
2. Line 104, it says “GBM33 displayed losses in chromosomes 9/10/19, including PTEN and MGMT…”. However, line 106, it says “…GBM33 exhibited wild-type IDH1/2 status and methylated MGMT.” Clarification needed explaining how MGMT is methylated if it is lost.
3. Consider providing side-by-side oncoscan array data for GBM33, LN-18, U87 and LN-229 (under same culture condition), which would be helpful to assess how GBM33 is different/unique from the other three cell lines.
4. Line 176, it says, “GBM33 was capable of producing cilia (Fig 4A), unlike LN-18 and LN-229 (Fig 4B abs C)”. However, the figure shows opposite results. Please explain. Also, ‘producing cilia’ is probably an inappropriate term; all the immunofluorescence results show is expression of alpha-tubulin. Electron microscopic results probably would be needed to demonstrate presence of actual cilia.
5. Line 215, it says “…we address this gap by describing the immortalization of three GBM cell lines and thoroughly characterizing them…”. Nowhere in the manuscript, there is derivation and thorough characterization of three cell lines but only one. Please clarify.
Also, ‘immortalization’ of cell lines in the lab is usually done by employing some standard bioengineering process, which was not done for GBM33.
Further, all GBM-derived cells are probably already immortalized without any requirement for immortalization. Consequently, in the context of the work presented in the manuscript, the word ‘immortalized’ is probably unnecessary.
6. I find the Introduction very well written.
Author Response
We would like to acknowledge the manuscript reviewers for their generous allocation of time and their insightful, constructive comments. Their evaluation and feedback contributed to the refinement of this work, improving the quality and clarity of our manuscript.
Please find bellow a point by point answer to the reviewer's comments:
- Line 93, it says, “…no staining for synaptophysin or GFAP”., whereas in line 125, it says “…low staining for the astrocytic marker GFAP”. Please explain why the tumor was negative for GFP whereas a cell line derived from it expressed GFAP at low level.
The original tumor tissue was evaluated by IHC in the diagnostic setting, performed in an automated fashion using pre-established conditions. To identify low levels of GFAP expression it is often needed to perform a new standardization of IHC conditions. The immunofliuorescence of the cells, on the other hand, was standardized using positive (astrocyte cells) and negative (fibroblast cells) expression controls and thus we were able to identify even lower patterns of GFAP expression.
- Line 104, it says “GBM33 displayed losses in chromosomes 9/10/19, including PTEN and MGMT…”. However, line 106, it says “…GBM33 exhibited wild-type IDH1/2 status and methylated MGMT.” Clarification needed explaining how MGMT is methylated if it is lost.
The CNV plot shown in Figure 1 is an estimate based on the data of the EPIC methylation array. It displays a very typical CNV GBM profile that includes the loss of one copy of chromosomes 9 and 10. The remaining copy of MGMT also presented promoter methylation suggesting a complete absence of expression of this gene. This information was added to the text.
3.Consider providing side-by-side oncoscan array data for GBM33, LN-18, U87 and LN-229 (under same culture condition), which would be helpful to assess how GBM33 is different/unique from the other three cell lines.
As requested, copy number variations from LN-18, LN-19 and U87 cells lines, obtained from the Cancer Cell Line Encyclopedia were added to Figure 2.
4.Line 176, it says, “GBM33 was capable of producing cilia (Fig 4A), unlike LN-18 and LN-229 (Fig 4B abs C)”. However, the figure shows opposite results. Please explain. Also, ‘producing cilia’ is probably an inappropriate term; all the immunofluorescence results show is expression of alpha-tubulin. Electron microscopic results probably would be needed to demonstrate presence of actual cilia.
Primary cilia were identified by the presence of acetylated alpha tubulin, a cytoskeleton component of these organelles. Immunolabeling for acetylated alpha tubulin is vastly accepted in the literature as a ciliary marker (e.g. Hua, K., Ferland, R.J. Fixation methods can differentially affect ciliary protein immunolabeling. Cilia 6, 5 (2017). https://doi.org/10.1186/s13630-017-0045-9).
Figure 4 depicts exactly what is described in the text, presence of cilia in GBM33 and absence in LN18 and LN229. The primary cilia are structures of approximately 1 to 10um long and are observed as one structure per cell. To facilitate the visualization for the unexperienced eye, we included arrowheads in Figure 4. The pattern of labeling observed in LN-18 and LN-229 does not correspond to cilia .
- Line 215, it says “…we address this gap by describing the immortalization of three GBM cell lines and thoroughly characterizing them…”. Nowhere in the manuscript, there is derivation and thorough characterization of three cell lines but only one. Please clarify.
We apologize for this inaccuracy and corrected the text accordingly.
- Also, ‘immortalization’ of cell lines in the lab is usually done by employing some standard bioengineering process, which was not done for GBM33. Further, all GBM-derived cells are probably already immortalized without any requirement for immortalization. Consequently, in the context of the work presented in the manuscript, the word ‘immortalized’ is probably unnecessary.
We agree with the reviewer and removed all mentions to immortalization in the text.
Reviewer 3 Report
Comments and Suggestions for Authors
Since I am not a molecular geneticist but biochemist and immunologist I will concentrate on my subjects. Investigated metabolic pathways were well examined, described and discussed also precise data on metylation and Copy Number Variation (CNV) of genome analysis seem to be well performed and presented. However, some important data or proper references concerning the experimental conditions that might affect the cell properties, are missing. Manuscript may be published after certain data supplementations and corrections as follow:
1).Very important issue is that no precise characteristics of patients from wchich samples of the tumor were taken and procedure ( biopsy or during the partial surgery?) were given, only: „We obtained one tumor fragment each from two different patients diagnosed with GBM at the AC Camargo Cancer Center in Sao Paulo, Brazil. The acquisition of the tumors was conducted with informed consent from the patients, and the study was approved by the Institutional Ethics Committee (Approval No. 082/18)”.
2).The information: „Patient GBM33 was a male diagnosed at the age of 76. MRI studies indicated the presence of a mass in the right frontal lobe (Figure 1A). The patient experienced gait alteration and dysarthria and underwent partial surgical removal of the tumor in 2016”, should be given in Materials and Methods. What about the other patient?
Also is no mentioned if patients were under some kind of treatment that might have the effect on cell properties.
3.). 4.1. Commercially Available Cells Lines, 4.2. Primary cell culture, 4.6. Western Blot,
No proper characteristic of cell culture conditions was given: culture dishes, cels density. Also more informations about „garamycin (Hipolabor Farmacêutica)” or proper reference should be given. It is important to remember, that gentamycin have antiproliferative effect; induce apoptosis and decrease the gene expression of neural stem cell markers (Elliott and Jiang .PLoS One. 2019;14(4):e0214586, Siti Nazihahasma andFarizan. Gulhane Med J. 2020;62:224-30).
4). 4.3. Immunofluorescence assay:were cell washed between antibody aplications and incubated with mixing?
5). 4.4. Soft agar colony formation assay: what was the final DMEM concentration?
6). 3. Discussion: Part of Discussion 187-210 belongs to Introduction. Discussion contains many repetitions from the Results instead concentration mainly on data from other researchers.
Comments on the Quality of English LanguageMinor editing of English language required
Author Response
We would like to acknowledge the manuscript reviewers for their generous allocation of time and their insightful, constructive comments. Their evaluation and feedback contributed to the refinement of this work, improving the quality and clarity of our manuscript.
Please find bellow a point by point answer to the reviewer's comments:
1) Very important issue is that no precise characteristics of patients from which samples of the tumor were taken and procedure ( biopsy or during the partial surgery?) were given, only: „We obtained one tumor fragment each from two different patients diagnosed with GBM at the AC Camargo Cancer Center in Sao Paulo, Brazil. The acquisition of the tumors was conducted with informed consent from the patients, and the study was approved by the Institutional Ethics Committee (Approval No. 082/18)”.
We included in the text information about sample obtention.
2) The information: Patient GBM33 was a male diagnosed at the age of 76. MRI studies indicated the presence of a mass in the right frontal lobe (Figure 1A). The patient experienced gait alteration and dysarthria and underwent partial surgical removal of the tumor in 2016”, should be given in Materials and Methods. What about the other patient? Also is no mentioned if patients were under some kind of treatment that might have the effect on cell properties.
Only one patient was included in the study. We apologize for the correct mention and corrected the text. The patients received no treatment prior surgery. We included this information in the text.
3.) 4.1. Commercially Available Cells Lines, 4.2. Primary cell culture, 4.6. Western Blot,
No proper characteristic of cell culture conditions was given: culture dishes, cels density. Also more informations about garamycin (Hipolabor Farmacêutica)” or proper reference should be given. It is important to remember, that gentamycin have antiproliferative effect; induce apoptosis and decrease the gene expression of neural stem cell markers (Elliott and Jiang .PLoS One. 2019;14(4):e0214586, Siti Nazihahasma andFarizan. Gulhane Med J. 2020;62:224-30).
Culture dishes and cell density information were added to the text as requested. Garamycin solution is supplied as a sterile 40mg/ml aqueous solution of Gentamycin Sulfate suitable for human use.
4). 4.3. Immunofluorescence assay: were cell washed between antibody aplications and incubated with mixing?
Cells were washed 3 times of 5 minutes each with PBS after every step of incubation. The information was added to the text. Incubations were not performed with mixing.
5). 4.4. Soft agar colony formation assay: what was the final DMEM concentration?
For soft agar colony formation assay DMEM was prepared as a 2x solution and mixed with sterile agarose in a 1:1 proportion to obtain a final concentration of 1x.
6). 3. Discussion: Part of Discussion 187-210 belongs to Introduction. Discussion contains many repetitions from the Results instead concentration mainly on data from other researchers.
As requested by the reviewers the first two paragraphs of the discussion were moved to the introduction section. Additional alterations in the discussion section were performed as requested by the other reviewers as well.
Reviewer 4 Report
Comments and Suggestions for Authors
In the manuscript: 'Establish and Comprehensive Molecular Characterization of an immortalized glioblastoma cell line from a Brazil Patient' the authors report the description of a new glioblastoma cell line: isolated from a patient biopsy: the GBM 33 line. Interesting but cannot be considered innovative; there are already other works that highlight the importance of primary glioblastoma lines from patient tissue Oliva M.A:, Staffieri S, Castaldo S, Giangaspero F, Esposito V, Arcella A. J Neurooncol.2021 Jan;151(2):123- 133.
in this paper the authors describe the genetic characteristics and the methylation profile of the new GBM 33 line satisfactorily, however I have some doubts:
1) I recommend implementing the introduction by clearly specifying the role of MGMT gene methylation in the response to chemotherapy and radiotherapy treatment.
2) Beyond the methylation profile, the authors highlighted the lack of expression of the protein
3) The authors compare GBM 33 cells with the commercial lines LN-229 and LN-18 but it is known that these lines have a different methylation profile, in particular LN-18 has the unmethylated MGMT promoter.
4) The distribution of the cilium is interesting and shows how the primary GBM 33 is more reactive to the stimuli of the commercial lines.
5) The authors discuss the data coming from the characterization of the primitives but give little importance to the description of new targets and new markers for personalized therapy of GBM.
Comments on the Quality of English LanguageModerate editing of English language required
Author Response
We would like to acknowledge the manuscript reviewers for their generous allocation of time and their insightful, constructive comments. Their evaluation and feedback contributed to the refinement of this work, improving the quality and clarity of our manuscript.
Please find bellow a point by point answer to the reviewer's comments:
1) I recommend implementing the introduction by clearly specifying the role of MGMT gene methylation in the response to chemotherapy and radiotherapy treatment.
The information was added to the text as requested.
2) Beyond the methylation profile, the authors highlighted the lack of expression of the protein
We assumed the reviewer is still referring to MGMT. If this is the case, we did not highlight the lack of expression of the protein. However, our data shows that one copy of MGMT is loss and the promoter in the other copy of the gene is methylated, suggesting absent expression of this enzyme.
3) The authors compare GBM 33 cells with the commercial lines LN-229 and LN-18 but it is known that these lines have a different methylation profile, in particular LN-18 has the unmethylated MGMT promoter.
We have compared the GBM33 cell line with cells presenting unmethylated (LN18) or methylated (U87 and LN229) MGMT promoter. It is known that sensitivity to temozolamide is highly affected by MGMT activity and in this sense, GBM33 sensitivity is equivalent to U87, which also presents methylated MGMT (Figure 3B).
4) The distribution of the cilium is interesting and shows how the primary GBM 33 is more reactive to the stimuli of the commercial lines.
As requested by reviewer 1, we improved the text on primary cilia on the results and discussion section.
5) The authors discuss the data coming from the characterization of the primitives but give little importance to the description of new targets and new markers for personalized therapy of GBM.
We appreciate the effort of the reviewer into giving constructive critics, however we found it a bit challenging to fully grasp the meaning of this question, as the sentences sounded a bit unclear. We assumed the referee would like us to highlight the description of new variants found that could potentially act as new targets and new markers for personalized therapy of GBM. We have thus added this information to the new Table 2.
Round 2
Reviewer 2 Report
Comments and Suggestions for Authors
Explanation/clarification/addition provided by the authors are accepted.